# WEE1 Inhibitor Adavosertib Exerts Antitumor Effects on Colorectal Cancer, Especially in Cases with *p53* Mutations

**DOI:** 10.3390/cancers16183136

**Published:** 2024-09-12

**Authors:** Misa Ariyoshi, Ryo Yuge, Yuki Kitadai, Daisuke Shimizu, Ryo Miyamoto, Ken Yamashita, Yuichi Hiyama, Hidehiko Takigawa, Yuji Urabe, Shiro Oka

**Affiliations:** Department of Gastroenterology, Hiroshima University Hospital, Hiroshima 734-0037, Japan; misa4235@hiroshima-u.ac.jp (M.A.); s025eb@hiroshima-u.ac.jp (Y.K.); dshimizu@hiroshima-u.ac.jp (D.S.); ryo4book@hiroshima-u.ac.jp (R.M.); kenyama5@hiroshima-u.ac.jp (K.Y.); yhiyama@hiroshima-u.ac.jp (Y.H.); hidehiko@hiroshima-u.ac.jp (H.T.); beyan13@hiroshima-u.ac.jp (Y.U.); oka4683@hiroshima-u.ac.jp (S.O.)

**Keywords:** WEE1 inhibitor, colorectal cancer, cell cycle, *p53* mutation, apoptosis, *KRAS* mutation, orthotopic transplantation

## Abstract

**Simple Summary:**

WEE1 negatively regulates the G2/M cell cycle checkpoint, allowing sufficient DNA damage repair before mitosis. Cancer cells often have a defective G1/S checkpoint and rely on the G2/M checkpoint for DNA damage repair. In this study, we investigated the association between WEE1 expression and clinicopathological features of colorectal cancer (CRC) using surgical specimens. Using various CRC cell lines with different mutation statuses, we compared the effects of the WEE1 inhibitor adavosertib on cell proliferation, cell cycle, and apoptosis induction, especially focusing on *p53* and *KRAS* mutations involved in the G1/S checkpoint. We also performed therapeutic experiments with adavosertib in an orthotopic transplanted CRC mouse model using two cell lines with different mutation statuses to evaluate its antitumor effect and impact on the tumor immune microenvironment. Our findings indicate that WEE1 inhibitors can be used for treating CRC, particularly in cases with *p53* mutations.

**Abstract:**

Inhibition of WEE1, a key regulator of the G2/M checkpoint of the cell cycle, induces apoptosis by initiating mitosis without repairing DNA damage. However, the effects of WEE1 inhibitors on the tumor immune microenvironment in colorectal cancer (CRC) remain unclear. Here, we investigated the association between WEE1 expression and CRC clinicopathological features using surgically resected CRC specimens and assessed the antitumor effects of a WEE1 inhibitor using CRC cell lines and orthotopic transplantation mouse models. WEE1 expression was not correlated with the clinicopathological features of CRC. The WEE1 inhibitor suppressed cell proliferation in a concentration-dependent manner in all CRC cell lines. It also increased the percentage of cells in the G2/M phase and apoptotic cells, especially in cell lines with *p53* mutations, but did not alter these cell percentages in most *p53* wild-type cell lines. In the orthotopic mouse model of CRC, tumor volume was significantly reduced in the WEE1 inhibitor-treated group compared to that in the control group. RNA sequencing and immunohistochemistry analyses of mouse tumors revealed that treatment with the WEE1 inhibitor activated tumor immunity and suppressed stromal reactions. These results demonstrate the potential antitumor effects of WEE1 inhibitors in CRC, particularly in patients with *p53* mutations.

## 1. Introduction

WEE1 is a nuclear kinase belonging to the Ser/Thr family and an important regulator of the G2/M phase checkpoint. WEE1 functions by phosphorylating 15 tyrosine residues of cyclin-dependent kinase 1 (CDK1), thereby inhibiting the activation of the cyclin B–CDK1 complex and the transition from the G2 phase to the M phase of the cell cycle [1,2].

Normal cells repair damaged DNA during G1 arrest; however, cancer cells often exhibit deficiencies in the G1/S checkpoint, making them reliant on the G2/M checkpoint for DNA repair. WEE1 regulates the initiation of mitosis and arrests the cell cycle in the G2 phase, allowing for DNA repair [3]. Inhibitors of WEE1, such as adavosertib, disrupt this regulatory function and facilitate the initiation of mitosis in cells with unrepaired DNA damage. This disruption leads to the accumulation of genomic instability and apoptosis.

*p53* and *KRAS* play crucial roles in cell cycle regulation. *p53* is a central regulator of the G1/S checkpoint [4], whereas *KRAS* mutations induce genomic instability by promoting the transition to the S phase and inducing cell division through sustained CDK2 activation [5,6]. Therefore, patients with mutations in *p53* or *KRAS* may be more sensitive to WEE1 inhibitors because of the dysfunctional G1/S checkpoint and increased dependence on the G2/M checkpoint for DNA repair. WEE1 inhibitors have been reported to enhance the efficacy of other chemotherapeutic agents such as 5-FU [7] and irinotecan [8] in colorectal cancer (CRC) with *p53* mutations. The efficacy of WEE1 inhibitors alone for treating metastatic CRC with *p53* and *KRAS* mutations has been demonstrated in a clinical trial [9]. Moreover, several other clinical trials have shown that WEE1 inhibitors, in combination with other agents, are effective against pancreatic cancer [10], non-small-cell lung cancer [11], and leukemia [12] with *KRAS* mutations. In contrast, in leukemia, lung cancer [13], and sarcoma [14], the efficacy of WEE1 inhibitors does not appear to be related to the presence of *p53* mutations. Furthermore, a few clinical trials have shown no response to WEE1 inhibitors as single agents in patients with solid tumors harboring *p53* mutations [15]. However, these reports did not directly compare differences in drug efficacy in the presence or absence of *p53* or *KRAS* mutations. Therefore, the precise effects of these genetic mutations on the antitumor efficacy of WEE1 inhibitors remain elusive.

WEE1 overexpression has been reported in various cancer types, including malignant melanoma [16], breast cancer [17], ovarian cancer [18], and glioma [19]. In addition, WEE1 expression has been reported to correlate with the prognosis of malignant melanoma and high-grade gliomas. In contrast, in non-small-cell lung cancer [20] and malignant melanoma [21], WEE1 is more highly expressed in low-grade lesions than in high-grade lesions, and the deletion of WEE1 has been reported to be an indicator of poor prognosis. Regarding WEE1 expression and prognosis in CRC, studies have shown increased metastases, disease progression, and poor prognosis in cases with high WEE1 expression [22], while others suggest no correlation with prognosis [23], rendering it a controversial topic. However, the prognostic impact of WEE1 expression and the presence of *p53* or *KRAS* mutations in CRC remains unclear.

Recently, the molecular effects and efficacy of WEE1 inhibitors were demonstrated in several types of carcinomas. Guo et al. [24] showed that the inhibition of WEE1 activates interferon pathways (the cGAS/STING and ERV/dsRNA pathways) to promote tumor immune responses in ovarian cancer and that combination therapy with WEE1 inhibitors and immune checkpoint inhibitors is effective. However, the effects of WEE1 inhibitors on cell proliferation, the cell cycle, and the tumor immune microenvironment in CRC are not yet fully understood.

Therefore, this study aimed to examine the effects of WEE1 expression, *p53* mutations, and *KRAS* mutations on the clinicopathological features and prognosis of CRC and the effects of WEE1 inhibitors in various CRC cell lines with different mutational statuses. Additionally, we investigated the effects of the WEE1 inhibitor on the tumor immune microenvironment using an orthotopic transplantation mouse model.

## 2. Materials and Methods

### 2.1. Reagents

The WEE1 inhibitor adavosertib (AZD1775) was purchased from MedChemExpress (Monmouth Junction, NJ, USA). The primary antibodies used were as follows: mouse monoclonal anti-WEE1 (B-11, sc-5285, Santa Cruz Biotechnology, Santa Cruz, CA, USA); human monoclonal anti-p53 antibody (DO-7, 413231, Nichirei Bioscience, Tokyo, Japan); anti-CD8α rabbit mAb (D4W2Z, 98941T, Cell Signaling Technology, Danvers, MA, USA); anti-CD4 rabbit mAb (D7D2Z, 25229S, Cell Signaling Technology); anti-FoxP3 rabbit mAb (D6O8R, 12653S, Cell Signaling Technology); anti-PD-L1 rabbit mAb (D5V3B, 64988, Cell Signaling Technology); and Ki-67 equivalent antibody (GTX16667, GeneTex, Irvine, CA, USA). Apoptosis was quantified using the terminal deoxynucleotidyl transferase dUTP nick-end labeling (TUNEL) assay with an ApopTag Plus Apoptosis Detection Kit (#S7100, Millipore-Chemicon International, Temecula, CA, USA).

### 2.2. Patients

Immunostaining was performed on paraffin-embedded tissue from resected specimens of 152 patients who underwent surgical resection for CRC at Hiroshima University Hospital between 2013 and 2015 and were diagnosed with *KRAS* and *BRAF* mutations and microsatellite instability (MSI) status. *KRAS* mutations include G12, G13D, and G13R mutations, which were evaluated collectively as *KRAS* mutations in the present study.

### 2.3. Immunohistochemistry for WEE1 Expression in Human CRC Tissue

To investigate the association between WEE1 expression and clinicopathological features, immunostaining for WEE1 was performed on 152 surgically resected CRC specimens. Formalin-fixed, paraffin-embedded tumor tissue was sectioned into 4 µm serial sections, and the extent of WEE1 expression was examined using immunostaining. Staining intensity scores for WEE1 expression were evaluated as follows: −, no staining; ±, weak; +, moderate; ++, strong reaction intensity. − and ± were defined as negative staining; + and ++ were defined as positive staining.

### 2.4. Classification of Human CRC Tissue According to the p53 Mutation Status

To examine the association between WEE1 expression and *p53* mutations, *p53* mutation status was examined using immunostaining in CRC tissue samples. Based on the distribution pattern of positive cells, the following classifications were made: diffuse strong, nuclei were darkly stained and clearly positive at 4× magnification; diffuse weak, nuclei were lightly stained and clearly positive at 10× magnification or more; nested, scattered clusters of positive cells with no negative cells mixed in; mosaic, diffuse presence of positive cells with mixed negative cells; sporadic, scattered presence of positive cells; negative, no positive cells. Furthermore, we classified diffuse strong, nested, and negative as *p53* mutations and mosaic, sporadic, and diffuse weak as *p53* wild types, according to previous reports [25,26].

### 2.5. WEE1 Expression Analysis Using the University of California, Santa Cruz (UCSC) Cancer Genomics Browser

UCSC Xena (https://xena.ucsc.edu/) is an analytical tool used to explore functional genomic datasets and to correlate genomic and phenotypic variables [27]. We used this platform to analyze WEE1 expression in colorectal cancer and evaluate its association with prognosis.

### 2.6. Cell Lines

The BALB/c mouse CRC cell line CT26 was obtained from the American Type Culture Collection (Manassas, VA, USA). The C57BL6 mouse-derived CRC cell line MC38 was obtained from Alstem Cell Advancement (Richmond, CA, USA). The human CRC cell lines DLD1, HCT116, LoVo, HT29, Caco2, and RKO were kindly provided by the Health Science Research Resources Bank (Osaka, Japan). KM12SM was provided by Dr. Isaiah J. Fidler (University of Texas, Austin, TX, USA).

### 2.7. RNA Extraction and Quantitative Polymerase Chain Reaction (PCR) Analysis of WEE1 Expression in CRC Cell Lines

Total RNA was extracted from CRC cell lines and mouse tumors using the RNeasy Mini Kit (Qiagen, Hilden, Germany) according to the manufacturer’s instructions. The first-strand cDNA synthesis kit (Amersham Biosciences, Piscataway, NJ, USA) was used to synthesize cDNA from 1 µg of total RNA. After reverse transcription of RNA to cDNA, a quantitative reverse transcription-polymerase chain reaction was performed using the LightCycler FastStart DNA Master SYBR Green I Kit (Roche Diagnostics, Basel, Switzerland), according to the manufacturer’s recommended protocol. The primers used in this study are shown in Table 1. The reactions were performed in triplicate. To correct for differences in RNA quality and quantity between samples, expression values were normalized to those of glyceraldehyde-3-phosphate dehydrogenase (GAPDH). The relative expression of WEE1 in each cell line was calculated according to the ΔΔCT method using normal human/mouse (BALB/c for CT26, C57BL6 for MC38) colon mucosa as the control. Normal human colonic mucosal RNA used as a control was extracted from existing formalin-fixed paraffin-embedded (FFPE) sections of clinical specimens (endoscopic biopsy tissue taken from the colonic mucosa of a healthy adult male) using the RNeasy FFPE Kit for RNA Extraction (Qiagen, Hilden, Germany) according to the protocol recommended by the manufacturer.

### 2.8. In Vitro Adavosertib Effects on Cancer Cell Proliferation

For in vitro experiments, cancer cells were cultured in RPMI-1640 medium with a 10% fetal bovine serum (Sigma-Aldrich, St. Louis, MO, USA) and 1% penicillin–streptomycin mixture at 37 °C in 5% CO_2_. The effect of the WEE1 inhibitor, adavosertib, on cell proliferation was evaluated in seven human CRC cell lines (DLD1, HCT116, LoVo, HT29, Caco2, KM12SM, and RKO) and two mouse-derived CRC cell lines (CT26 and MC38). CRC cells (5 × 10^4^ cells per well) were seeded into 24-well plates (Essen ImageLock; Essen Bioscience, Ann Arbor, MI, USA). The cells were treated with various concentrations of adavosertib (0, 200, 500, and 1000 nM). Growth curves were generated from bright-field images obtained using a label-free, high-content, timelapse assay system (IncuCyte Zoom; Essen Bioscience). All experiments were performed in quadruplicate.

### 2.9. In Vitro Adavosertib Effects on Cancer Cell Cycle

The effect of adavosertib on the cell cycle was evaluated in seven human CRC cell lines (DLD1, HCT116, LoVo, HT29, Caco2, KM12SM, and RKO) and two mouse CRC cell lines (CT26 and MC38). Cells collected at 0, 4, 12, and 24 h after exposure to 1000 nM adavosertib were classified into G1, S, and G2/M phases by flow cytometry according to the Cell Cycle Assay Solution Blue protocol (Dojindo, Kumamoto, Japan), and the percentage of cells in each phase was examined over time. All experiments were performed in triplicate.

### 2.10. In Vitro Adavosertib Effects on Cancer Cell Apoptosis

The effects of adavosertib on the induction of apoptosis in seven human CRC cell lines (DLD1, HCT116, LoVo, HT29, Caco2, KM12SM, and RKO) and two mouse CRC cell lines (CT26 and MC38) were evaluated. Cells collected at 0, 6, 24, and 48 h after exposure to 1000 nM adavosertib were classified as apoptotic, live, or necrotic by flow cytometry according to the protocol of the Annexin V-FITC Apoptosis Detection Kit (Nacalai Tesque, Kyoto, Japan), and the degree of increase in the percentage of apoptotic cells over time was evaluated. This kit was set up by adding Annexin V binding solution and propidium iodide (PI) solution to cell suspensions and letting them react, followed by flow cytometry to classify Annexin V (+) and PI (−) as apoptosis, Annexin V (+) and PI (+) as necrosis, and Annexin V (−) and PI (−) as viable cells. All experiments were performed in triplicate.

### 2.11. Establishment of a CRC Orthotopic Transplantation Mouse Model and Adavosertib Treatment

Female C57BL6 mice and BALB/c mice (both were six weeks old) were obtained from Jackson Laboratory Japan, Inc. (Yokohama, Japan). The mice were maintained under specific pathogen-free conditions and used at nine weeks of age.

Animal experiments were performed using the CRC cell line MC38, derived from C57BL6 mice. We prepared 2 × 10^6^ MC38 cells (stably expressing firefly luciferase; MC38-Luc) in 25 µL of Hank’s balanced salt solution and transplanted them orthotopically into the cecum wall of C57BL6 mice to generate a mouse model of CRC. Mice were divided into two groups: a control group receiving daily oral administration of methylcellulose and a treatment group receiving daily oral administration of 50 mg/kg/day adavosertib. Oral administration was initiated 14 days after tumor transplantation and was administered daily for 21 days. Thirty-five days after transplantation, the mice were euthanized, tumors were removed, and tumor volumes were measured.

We also performed animal experiments using the CRC cell line CT26. We prepared 5 × 10^4^ CT26 cells in 25 µL of Hank’s balanced salt solution and transplanted them orthotopically into the cecum wall of BALB/c mice to generate a mouse model of CRC. As with MC38, the mice were administered for 21 days, but dosing was started on day 3, and the mice were euthanized on day 24.

Tumor volume was calculated as V = (W^2^ × L)/2 (V: volume, W: short diameter, L: long diameter). Tumor tissues were fixed with a formalin-free immunohistochemistry zinc fixative provided as a ready-to-use solution (BD Pharmingen; BD Biosciences, Milpitas, CA, USA), paraffin-embedded, cut into 4 mm serial sections, and immunostained to evaluate the effect of adavosertib on the tumor immune microenvironment.

### 2.12. Immunofluorescence Staining of Mouse Tumor Specimens

Fluorescence immunostaining was performed for CD8α, CD4, FoxP3, and PD-L1 using the OPAL 4-Color Manual IHC Kit (Perkin Elmer, Norwalk, CT, USA). An all-in-one fluorescence microscope (BZ-X710, Keyence, Osaka, Japan) was used for confocal fluorescence imaging. For each specimen, five different microscopic fields of view were imaged at 200× magnification. All micrographs were obtained under the same conditions (exposure time, gain, illumination light intensity, and aperture stop). CD8−, CD4−, and FoxP3-positive cell counts were determined using the BZ-H3C hybrid cell count application in the BZX analysis software, version 1.3.1.1 (Keyence). The average cell count of the five microscopic fields of view of each sample was calculated.

### 2.13. Immunohistochemistry Staining of Mouse Tumor Specimens

Immunostaining with diaminobenzidine (DAB) was performed for Ki67 and CD8 staining. To detect apoptotic cells, TUNEL staining was performed using an ApopTag Plus Apoptosis Detection Kit (#S7100, Millipore-Chemicon International, Temecula, CA, USA) according to the manufacturer’s protocol. The Ki-67 labeling index and apoptotic index were calculated by taking images of five different microscopic fields of view at 400× magnification, counting the number of Ki-67 and TUNEL stain-positive cells in tumor cells and expressing them as percentages.

### 2.14. RNA Sequencing

We performed RNA sequencing on the tumor specimens from an orthotopic transplanted mouse model using MC38. MC38 was selected because in vitro studies showed that MC38 had a stronger effect on the cell cycle and apoptosis caused by adavosertib, so we decided that MC38 was more promising for tumor shrinkage and for evaluating the effect of adavosertib on tumors. Tumors from the treatment and control groups were removed and mechanically crushed using a homogenizer. Total RNA was extracted from tissue homogenates using the Qiagen RNeasy Mini Kit according to the manufacturer’s protocol. Library construction and data processing were performed at Beijing Genome Institute (Beijing, China). The libraries were sequenced using the DNBSEQ-G400RS platform to obtain high-quality reads. Sequence alignment was performed using the GRCm38 mouse reference genome version GCF_000001635.26_GRCm38.p6 (https://www.ncbi.nlm.nih.gov/assembly/GCF_000001635.26 (accessed on 10 March 2023)).

### 2.15. Functional Enrichment Analysis and Gene Set Enrichment Analysis (GSEA)

Dr. Tom multiple omics data mining system (https://biosys.bgi.com; Beijing Genome Institute, Beijing, China) was used to identify relevant differentially expressed genes (DEGs) and performed Gene Ontology (GO) and Kyoto Encyclopedia of Genes and Genomes (KEGG) pathway enrichment analyses and GSEA; *p*-values were calculated, false discovery rate-corrected *p*-values were used to obtain q-values, and q-values < 0.25 were considered to indicate significant enrichment.

### 2.16. Statistical Analysis

All statistical analyses were conducted using the EZR software (version 1.60, Saitama Medical Centre, Jichi Medical University, Saitama, Japan) [28]. Clinicopathological characteristics were analyzed using the χ^2^ test for the comparison of categorical data and Welch’s *t*-test for the comparison of continuous data. In multiple comparisons for continuous data, one-way analysis of variance (ANOVA) followed by Dunnett’s post hoc test was conducted. Kaplan–Meier curves and overall survival were analyzed using the log-rank test. Statistical significance was set at *p* < 0.05.

## 3. Results

### 3.1. WEE1 Expression in Surgical CRC Specimens Is Not Correlated with CRC Clinicopathological Features

We performed immunostaining of resection specimens from 152 patients who underwent surgery for CRC at Hiroshima University Hospital and were diagnosed with *KRAS* or *BRAF* mutations or MSI status. The patients were classified into positive (*n* = 71) or negative (*n* = 81) groups depending on the degree of WEE1 expression (Figure 1a). Analysis of the association between WEE1 expression in CRC specimens and clinicopathological features of CRC revealed no significant differences in age, sex, location, histological type, or stage between the WEE1-positive and -negative groups (Table 2).

Table 3 shows the association between WEE1 expression and genetic mutations. As shown in Figure 1b, *p53* was classified as wild type or mutant based on the distribution of immunostained positive cells. *BRAF* and MSI-high mutations were significantly more common (*p* < 0.05) in the WEE1-positive group, and the percentages of *KRAS* and *p53* mutations were not significantly different between the two groups.

### 3.2. Patients with WEE1-Positive and KRAS-Mutated CRC Have Poor Prognosis

Analysis of the association between WEE1 expression and overall survival (OS) in the TCGA database (GDC, TCGA, Colon Adenocarcinoma (COAD), based on the analysis of 487 samples) using Xena showed no significant difference in OS between the high and low WEE1 expression groups (log-rank test; *p* = 0.413; Figure 1c). Similarly, analysis using the 152 surgical cases did not reveal significant differences in OS between the WEE1-positive and -negative groups (log-rank test; *p* = 0.261; Figure 1d). Further classification by mutation status revealed that patients with positive WEE1 expression and *KRAS* mutations had significantly lower OS rates than those with negative WEE1 expression and wild-type *KRAS* (Figure 1e).

### 3.3. WEE1 Expression Tended to Be Higher in CRC Cell Lines than in Normal Colon Mucosa

Relative WEE1 expression levels when the control (normal colon mucosa) was set as 1 was determined in seven human CRC cell lines (DLD1, KM12SM, Caco2, HT29, HCT116, LoVo, and RKO), one BALB/c mouse-derived CRC cell line (CT26), and one C57BL6 mouse-derived colon cancer cell line (MC38) using real-time PCR (Figure 2a). WEE1 expression levels tended to be higher in all human CRC cell lines, with the exception of LoVo, than in normal colon mucosa. Additionally, in both mouse colon cancer cell lines (CT26 and MC38), the expression level of WEE1 tended to be higher than that in normal colonic mucosa. The gene mutation statuses *(p53*, *KRAS*, and *BRAF*) and MSI of each cell line are shown in Table 4 [29,30]. The findings revealed no association between the gene mutation status and WEE1 expression levels (Table 4).

### 3.4. WEE1 Inhibitor Suppresses CRC Cell Line Proliferation in a Concentration-Dependent Manner

Next, we analyzed the effect of the WEE1 inhibitor on the proliferation of all the cell lines (seven human and two mouse cell lines, Section 2.6) using the timelapse system. Treatment of the cell lines with varying concentrations of adavosertib, a WEE1 inhibitor, and measurement of the area of the cancer cell-occupied region using IncuCyte Zoom (Ver. 2018A) revealed cell proliferation inhibition in a concentration-dependent manner, regardless of the degree of WEE1 expression or the presence or absence of genetic mutations (Figure 2b). All cell lines showed sufficient proliferation inhibition at 1000 nM; however, the degree of proliferation inhibition at 500 nM varied among the cell lines. Furthermore, no consistent trend was observed in changes in WEE1 expression levels, gene mutation status, or effective drug concentration.

### 3.5. WEE1 Inhibitor Increases the G2/M-Phase Cell Percentage, Especially in KRAS-Mutated, p53-Mutated Cell Lines

We evaluated the effects of the WEE1 inhibitor on the cell cycle of CRC cell lines. Seven human colon cancer cell lines (DLD1, KM12SM, Caco2, HT29, HCT116, LoVo, and RKO) and two mouse CRC cell lines (CT26 and MC38) were treated with 1000 nM adavosertib, and cell cycle analysis was performed to classify cells in the G1, S, and G2/M phases using flow cytometry (Figure 3a). In DLD1 harboring both *KRAS* and *p53* mutations, the percentage of cells in the G2/M phase increased significantly after addition of WEE1 inhibitor. In cell lines with *p53* mutations and wild-type *KRAS*, KM12SM showed an increased percentage of cells in G2/M phase, while Caco2 and HT29 showed little change in cell cycle. Of the *p53* mutant and *KRAS* wild-type cell lines, LoVo had fewer cell cycle changes, and HCT116 had an increased percentage of cells in G2/M phase, but this increase was less than that in DLD1 and KM12SM. RKO, which is the wild type for both *KRAS* and *p53*, showed no significant change in the percentage of cells in the G2/M phase. In CRC cell lines derived from mice, the percentage of cells in the G2/M phase increased in both CT26 and MC38 cells, especially in MC38 cells. CT26 harbors wild-type *p53* and *KRAS* mutations, and MC38 harbors wild-type *KRAS* and *p53* mutations. The findings indicates that the cell cycle was more variable in MC38 and was similar to the cell cycle analysis results for human-derived cell lines. These results indicate that the increase in the percentage of cells in the G2/M phase by the WEE1 inhibitor was partially dependent on the mutational status of *p53* and *KRAS*.

### 3.6. WEE1 Inhibitor Increases Apoptosis in CRC Cell Lines, except in KRAS-Mutant, p53 Wild-Type Cell Lines

Next, we assessed the changes in the percentage of apoptotic, viable, and necrotic cells over time in seven human colon cancer cell lines (DLD1, KM12SM, Caco2, HT29, HCT116, LoVo, and RKO) and two mouse colon cancer cell lines (CT26 and MC38). The cells were either untreated (0 h) or treated with 1000 nM adavosertib (6, 24, and 48 h) and classified into apoptotic, viable, and necrotic by flow cytometry using the Annexin V-FITC apoptosis detection kit (Figure 3b). All cell lines showed a time-dependent increase in the percentage of apoptotic cells, except for the cells harboring wild-type *p53* and *KRAS* mutations.

### 3.7. WEE1 Inhibitor Alone Induces Tumor Shrinkage, CD8-Positive T Cell Tumor Infiltration, and Tumor Cell Apoptosis in In Vivo CRC Model

To establish an orthotopic transplantation mouse model, MC38 cells (a CRC cell line derived from C57BL/6 mice) at 2.0 × 10^6^ were transplanted orthotopically into the cecum of C57BL/6 mice. MC38 is a cell line with a *p53*-mutant and *KRAS* wild-type pattern, and in vitro experiments showed that adavosertib induced cell cycle changes and apoptosis. The mice were orally administered either 0.5% methylcellulose (control group; *n* = 5) or 50 mg/kg/day adavosertib (treatment group; *n* = 4) for 21 days, starting 14 days after transplantation at the same site. On day 35, the mice were euthanized to collect the tumors (Figure 4a). Tumor size was significantly smaller in the treatment group (median [range], 142 [63–288] mm^3^) than in the control group (1040 [365–1584] mm^3^; *p* < 0.01; Figure 4b,c).

To establish an orthotopic transplantation mouse model, CT26 cells (a CRC cell line derived from BALB/c mice) at 5.0 × 10^4^ were transplanted orthotopically into the cecum of BALB/c mice. CT26 is a cell line with a *KRAS*-mutant and *p53* wild-type pattern in which apoptosis was not well induced by adavosertib in in vitro experiments. The mice were orally administered either 0.5% methylcellulose (control group; *n* = 6) or 50 mg/kg/day adavosertib (treatment group; *n* = 6) for 21 days, starting 3 days after transplantation at the same site. On day 24, the mice were euthanized to collect the tumors (Figure 4d). Tumor volume comparisons showed a trend toward smaller tumor volumes in the treatment group (median [range], 325 [200–700] mm^3^) than in the control group (595 [250–800] mm^3^), but the difference was not as great as in MC38 and was not significant (Figure 4e,f).

Next, we performed immunostaining using the transplanted tumor specimens to evaluate the effect of the WEE1 inhibitor on the tumor immune microenvironment. In the treatment group, CD8-positive T cell infiltration into the tumor was significantly increased compared to that in the control group (Figure 5a,e). In contrast, the infiltration of CD4 and FoxP3 did not differ significantly between the control and treatment groups (Figure 5b–e). Moreover, the expression of programmed cell death ligand 1 (PD-L1) was increased (Figure 5d), while Ki67 expression was significantly decreased in the treatment group compared to that in the control group, suggesting that the WEE1 inhibitor suppressed tumor cell proliferation (Figure 5f,h). TUNEL staining showed a significant increase in the number of positive cells in the tumors of the treatment group (Figure 5g,h), suggesting that the WEE1 inhibitor caused apoptosis of the tumor cells.

Transplanted tumor specimens from a mouse model of CRC using CT26 were also evaluated by immunostaining (Appendix A). The results showed that CD8-positive cells in the tumors significantly increased in the treatment group and Ki67-positive cells significantly decreased in the treatment group, suggesting that adavosertib induced the infiltration of CD8-positive T cells into the tumors and suppressed tumor growth. However, unlike MC38, TUNEL staining showed no difference in the percentage of positive cells between the treatment and control groups, indicating that apoptosis was not induced by adavosertib, consistent with the in vitro experimental results.

### 3.8. Comprehensive Gene Expression Analysis of Transplanted Tumors Using RNA Sequencing

Subsequently, RNA sequencing was performed on tumor specimens from an orthotopic transplanted mouse model of CRC using MC38, which is thought to be more sensitive to WEE1 inhibitors based on previous results, to assess the effects of adavosertib treatment on the gene expression profiles of the orthotopic transplantation mouse models. Significant DEGs with *p*-values < 0.05 and fold changes greater than 2 were identified in both groups. The analysis revealed that 77 genes were upregulated and 137 were downregulated in the treatment group compared to those in the control group (Figure 6a).

KEGG pathway analysis showed that treatment with adavosertib significantly upregulated the IL-17, TNF, chemokine, JAK-STAT, and cytosolic-DNA sensing pathways and strongly downregulated the PI3K-AKT signaling, HIF-1 signaling, and stromal activation pathways (Figure 6b). Additionally, GO analysis showed that adavosertib treatment upregulated genes related to the response to interferon (especially interferon-gamma), T cell migration and activation, immune response, apoptosis, and DNA damage response, and downregulated those related to stromal activation, response to hypoxia, chromatin remodeling, angiogenesis, cell proliferation, and regulatory T cells. Regarding the cell cycle, the transition from G0 to G1 was upregulated, whereas the transition from G1 to G0 was downregulated (Figure 6c). These results indicate that the WEE1 inhibitor induced cell cycle progression and apoptosis.

Subsequently, GSEA revealed upregulation of the G2/M checkpoint and G1/S transition. In addition, genes associated with various stages of mitosis in the M phase, such as condensation of prophase chromosomes, resolution of sister chromatid cohesion in prometaphase, amplification of signals from the kinetochore, separation of sister chromatids, and activation of APC/C (anaphase-promoting complex/cyclosome) involved in the transition from metaphase to anaphase chromosomes, were upregulated. These findings indicate that WEE1 inhibition promotes mitotic progression. Additionally, the genes associated with the S phase, including those related to DNA replication and the DNA double-strand break response at the intra-S checkpoint, were upregulated. Genes linked to stromal activation were also strongly downregulated (Figure 6d).

## 4. Discussion

While the number of CRC cases detected and treated at an early stage is increasing, CRC remains one of the leading causes of cancer-related mortality, with many cases having a poor prognosis. Immune checkpoint inhibitors, which have gained considerable attention in recent years, has been shown to be effective for MSI-high cases, but their effectiveness in MSS cases, which make up the majority of CRC cases, is poor, highlighting the need for more effective drug therapies. In this study, we evaluated the efficacy of adavosertib, which targets WEE1 kinase, which plays an important role in cell cycle regulation and DNA damage identification and repair, against CRC.

In this study, using surgical CRC specimens, no significant association was found between WEE1 expression and *p53* or *KRAS* mutations; however, WEE1-positive cases were significantly more likely to have *BRAF* mutations. In a report on melanoma, the inhibition of ^V600E^*BRAF* decreased WEE1 expression, suggesting that WEE1 is located downstream of ^V600E^ BRAF in the MAPK signaling cascade [31] and that mutations in BRAF activate the MAPK pathway and increase WEE1 expression. Additionally, significantly more MSI-high cases were observed in WEE1-positive patients. This could be attributed to the insufficient DNA damage repair in MSI-high cells because of a defective mismatch repair mechanism during DNA replication, resulting in increased expression of WEE1, a regulator of the G2/M checkpoint.

Furthermore, our findings showed no significant association between WEE1 expression and prognosis. Inhibiting a target molecule may be beneficial when its expression level correlates with prognosis. However, some molecules can still be therapeutic targets even without a significant correlation between their expression level and prognosis. WEE1 inhibitors have shown efficacy in a phase 2 clinical trial in CRC [9]. The mechanism inferred from the results of this study is that depletion of WEE1 in cancer cells by WEE1 inhibitors promotes the cell cycle, despite the presence of DNA damage, and induces apoptosis, which may have an antitumor effect. In contrast, cases with *KRAS* mutation tend to have a worse prognosis, consistent with previous reports [32]. In addition to *KRAS* mutations, patients with high WEE1 expression had a significantly poorer prognosis. Thus, *KRAS* mutation alone affects OS, but it is possible that high expression of WEE1 may further affect the prognosis.

In vitro studies have shown that the presence of *p53* mutations strongly affects sensitivity to WEE1 inhibitors. WEE1 acts on the G2/M checkpoint and inhibits progression from the G2 to M phase. It also regulates the termination of the M phase, and increased WEE1 expression leads to the termination of the M phase. Therefore, the inhibition of WEE1 induces apoptosis through the early initiation of mitosis, resulting in a decrease in the number of cells in the G1 phase and a relative increase in the proportion of cells in the G2/M phase, as well as an extension of the M phase, increasing the percentage of G2/M phase cells. In the present study, cell cycle analyses revealed higher cell cycle changes in cell lines with *p53* mutations, particularly in those harboring both *p53* and *KRAS* mutations. However, HT29, a *p53* mutant cell line, did not show pronounced increase in G2/M phase cells. Cell cycle analysis showed a large increase in G2/M phase cells in cell lines in which both *p53* and *KRAS* were mutant, whereas in cell lines in which either *p53* or *KRAS* was mutant, some cell lines showed an increase in G2/M phase cells while others showed little change. There is no consistent trend as to which changes are more significant, and it is possible that factors other than *p53* and *KRAS* mutation status are associated with changes in the cell cycle, but this was beyond the scope of this study. On the other hand, in the apoptosis assay, WEE1 inhibitors tended to increase apoptotic cells in *p53* mutant and *KRAS* wild-type cell lines compared to cell lines with both *p53* and *KRAS* mutations. In this study, HT29 showed an increase in apoptotic cells, which is a reasonable result. In contrast, apoptotic cells were not increased in *KRAS* mutant and *p53* wild-type cell lines. A previous study on non-small-cell lung cancer with *KRAS* mutations reported that induction of DNA damage and apoptosis by WEE1 inhibitors occurs only in the presence of *p53* mutations because the PI3K–AKT pathway is inhibited by WEE1 inhibitors only in the presence of *p53* mutations [33]. Overactivation of AKT suppresses the DNA damage response, inhibits the apoptosis-inducing pathway, and promotes cell cycle progression, allowing cells with unrepaired DNA damage to continue dividing without being removed by apoptosis [34]. The PI3K–AKT pathway is located downstream of RAS and is activated in the presence of *KRAS* mutations. Therefore, in cell lines with *KRAS* mutations and wild-type *p53*, WEE1 inhibition did not induce apoptosis due to activation of the PI3K–AKT pathway. In contrast, in cell lines with both *KRAS* and *p53* mutations, WEE1 inhibition suppresses the AKT pathway and induces apoptosis, suggesting the potential effectiveness of WEE1 inhibitors in this context.

In our study of the effect of adavosertib on cell proliferation, all cell lines showed sufficient proliferation inhibition at 1000 nM. In a phase I clinical study of adavosertib (https://www.astrazenecaclinicaltrials.com/study/D6014C00005/ (accessed on 10 May 2024)), Cmax was 965.9 nM at a clinical dose of 300 mg/day, and therefore, 1000 nM was considered appropriate.

Among the human CRC cell lines, KM12SM, with a relatively low expression of WEE1, showed a concentration-dependent inhibition of cell proliferation, an increase in G2/M phase cells in the cell cycle analysis, and an increase in apoptotic cells. Conversely, RKO, which had relatively high WEE1 expression, showed the same inhibition of cell proliferation as the other cell lines in the proliferation assay, but with smaller changes in the cell cycle and a smaller increase in apoptotic cells. LoVo, which had the lowest expression of WEE1, showed only a small change in the cell cycle and no increase in apoptotic cells. However, this result may have been influenced by the mutation status of *KRAS*-mutant and *p53* wild-type cells. Based on these results, we did not find an obvious correlation between the degree of WEE1 expression and the effect of the WEE1 inhibitor.

In treatment experiments using the orthotopic transplanted mouse model of CRC, the WEE1 inhibitor significantly reduced tumor volume, and immunostaining revealed the infiltration of CD8-positive T cells into the tumors in the treated group. WEE1 inhibitors have been reported to activate the cytosolic-DNA sensing, interferon, and JAK-STAT pathways, thereby enhancing tumor immunity through CD8+ T cell infiltration, increased cytokine and chemokine signaling, and increased PD-L1 expression [24,35]. RNA sequencing of transplanted tumors in this study demonstrated that the WEE1 inhibitor strongly enhanced T cell proliferation, activation, and infiltration. It upregulated not only CD8-positive T cells but also helper T cells, such as Th1 and Th17 (CD4-positive T cells), and suppressed regulatory T cells. Moreover, a group of genes related to the stroma were significantly downregulated. Stromal components have been reported to act as barriers to immune cell infiltration in various carcinomas [36,37,38]. Therefore, stromal suppression by WEE1 inhibitors may play a role in the induction of immune cell infiltration. 

The RNA sequencing results effectively reflected the effects of WEE1 inhibition on each phase of the cell cycle. GO analysis revealed upregulation of the G0 to G1 transition, downregulation of the G1 to G0 transition, and exit from mitosis. GSEA showed upregulation of genes associated with various stages of mitosis in the M phase, indicating that WEE1 inhibition promotes cell cycle progression and prolongs the M phase. These findings of the present study align with those of a few recent studies, which reported that WEE1 is involved in controlling the termination of the M phase and inhibiting its progression to the M phase at the G2/M checkpoint [39,40]. During the cell cycle, WEE1 expression increases from the S to the G2 phase. At the G2/M transition, WEE1 is phosphorylated by kinases such as CDK1, Polo-like kinase 1 (PLK1), and AKT, leading to its decreased expression and transition to the M phase. Following this, mitosis advances as FCP1 dephosphorylates T239 of WEE1, leading to an increase in WEE1 activity, ultimately culminating in the conclusion of the M phase. Therefore, WEE1 inhibition has been shown to promote the transition to the M phase and the progression of mitosis while inhibiting the termination of the M phase, resulting in M phase prolongation [41].

The transition from the G1 phase to the S phase was also upregulated in GSEA. As the transition from the G1 phase to the S phase occurs when Cyclin E1 and CDK2 form a complex and phosphorylate the Rb protein, the inhibition of WEE1, which inhibits CDK2, may promote the transition from the G1 to S phase. 

Moreover, in the GO and GSEA analyses, the DNA damage response and detection of DNA damage at each checkpoint were upregulated by adavosertib, indicating that DNA damage was induced by WEE1 inhibition. This finding is consistent with those of previous studies [42,43] showing that WEE1 regulates the DNA replication rate by suppressing CDK2 activity at the intra-S checkpoint, thereby preventing replication from being initiated from numerous replication origins, resulting in excessive replication fork breaks and increased genomic instability.

In GSEA, genes related to the response to hypoxia were strongly downregulated by WEE1 inhibition, and KEGG pathway analysis showed downregulation of the HIF-1 signaling pathway. Hypoxia has been reported to enhance WEE1 expression [44] and induce DNA double-strand breaks, leading to cell cycle arrest at the G2 phase and activation of the DNA damage response (DDR) pathway [45]. These findings suggest that WEE1 plays an important role in the tumor response to hypoxia. Therefore, inhibiting WEE1 may suppress the hypoxic response, HIF-1 signaling pathway, and angiogenesis.

The association between the effects of WEE1 inhibitors and *p53* or *KRAS* mutation status has been investigated in various cancers; however, some studies have highlighted the correlation between *p53* or *KRAS* mutation status and the effects of WEE1 inhibitors [7], whereas others have not [13]. The results are controversial, and there are no reports of systematic analysis of clinical data and in vitro and in vivo experiments. In particular, few reports have examined this for CRC, and the only report on the efficacy of WEE1 inhibitors as single agents is the report on the clinical trial [9], which did not examine the difference in the efficacy of WEE1 inhibitors depending on the presence or absence of *p53* or *KRAS* mutations. Here, through an in vitro study using CRC cell lines of various mutational statuses, and an in vivo study using an orthotopic transplanted mouse model with two cell lines with different mutation statuses, we were able to compare the effects of WEE1 inhibitors on cell lines and transplanted tumor specimens. Therefore, we consider that this is a novel study. In this study, we found that WEE1 inhibitors suppressed the stromal reaction, but the detailed mechanism of this suppression is unknown. WEE1 inhibition has also been reported to affect glutamine metabolism, with reports that inhibition of WEE1 suppresses the glycolytic system, increasing dependence on glutamine metabolism [46] and that oncogenic KRAS regulates cell cycle checkpoints and glutamine metabolism [47]. Glutamine metabolism affects the tumor stroma, and it has been reported that the glutamine-metabolizing enzyme GLS1 promotes cancer growth via autophagy in the tumor stroma [48], and it is possible that the effect of WEE1 inhibitors on glutamine metabolism is involved in the suppression of the stromal reaction. We believe that pursuing the mechanism is a future issue.

In this study, we were unable to examine in detail the side effects of adavosertib. Previous reports [49,50] have suggested that WEE1 inhibitors may act in a cancer-specific manner. In this treatment experiment using an orthotopic transplanted mouse model, there was no significant decrease in body weight in the treatment group, indicating no serious side effects.

This study has several limitations. Firstly, in the experiments using clinical specimens, the mutation status of *p53* was determined by immunostaining, which may not always align with the actual *p53* mutation status, potentially affecting the accuracy of the evaluation. Secondly, while in vitro evaluation was possible using cell lines with various mutation statuses, in vivo evaluation was limited to MC38 cells with *p53* mutations and wild-type *KRAS* and CT26 cells with *KRAS* mutations and wild-type *p53*. In this study, we wanted to evaluate changes in the tumor immune microenvironment induced by WEE1 inhibition, so it was necessary to use a model of allogeneic immune response. Therefore, we did not perform animal experiments in which human-derived cell lines were transplanted into nude mice. Third, siRNA targeting was technically difficult at our laboratory and could not be performed. Fourth, in the in vivo experiment, the control and treatment groups were compared in terms of tumor volume, but since this was calculated using an approximate formula, there may have been a potential error. A comparison using tumor weight would have been more rigorous. In addition, as previously reported, MC38 often fails to grow in orthotopic or subcutaneously transplanted mouse models [51], and in this study, some individuals did not develop tumors after orthotopic transplantation or died during the treatment experiment, resulting in different and smaller group sizes. 

The results of this study indicate that in CRC with *p53* mutations, WEE1 inhibitors induce cell cycle changes, apoptosis, and suppressed cell proliferation in vitro and in vivo; these effects were followed by activation of tumor immunity and suppression of the stromal reaction, cell cycle progression and prolongation of the M phase, DDR, response to hypoxia, and inhibition of angiogenesis. These findings provide compelling evidence that personalized medicine using WEE1 inhibitors may be a viable approach for treating CRC with *p53* mutations. However, further studies are required to determine the characteristics of suitable patients for treatment.

## 5. Conclusions

Inhibition of WEE1 exerts antitumor effects against CRC by stimulating cell cycle progression, inducing apoptosis, enhancing tumor immunity, and suppressing the stromal response. In the case of both *p53* and *KRAS* mutants, WEE1 inhibitors caused significant cell cycle fluctuations and induced apoptosis. For *p53* mutant and *KRAS* wild type, apoptosis was more easily induced, the cell cycle was more variable, and antitumor effects were demonstrated in vivo. Conversely, for *p53* wild-type and *KRAS* mutant, apoptosis was less likely to be induced, and in vivo results also suggest that the antitumor effect of WEE1 inhibitors is less promising than in the case of *p53* mutant. Patients with *p53* mutations may benefit more from WEE1 inhibitors.

## Figures and Tables

**Figure 1 cancers-16-03136-f001:**
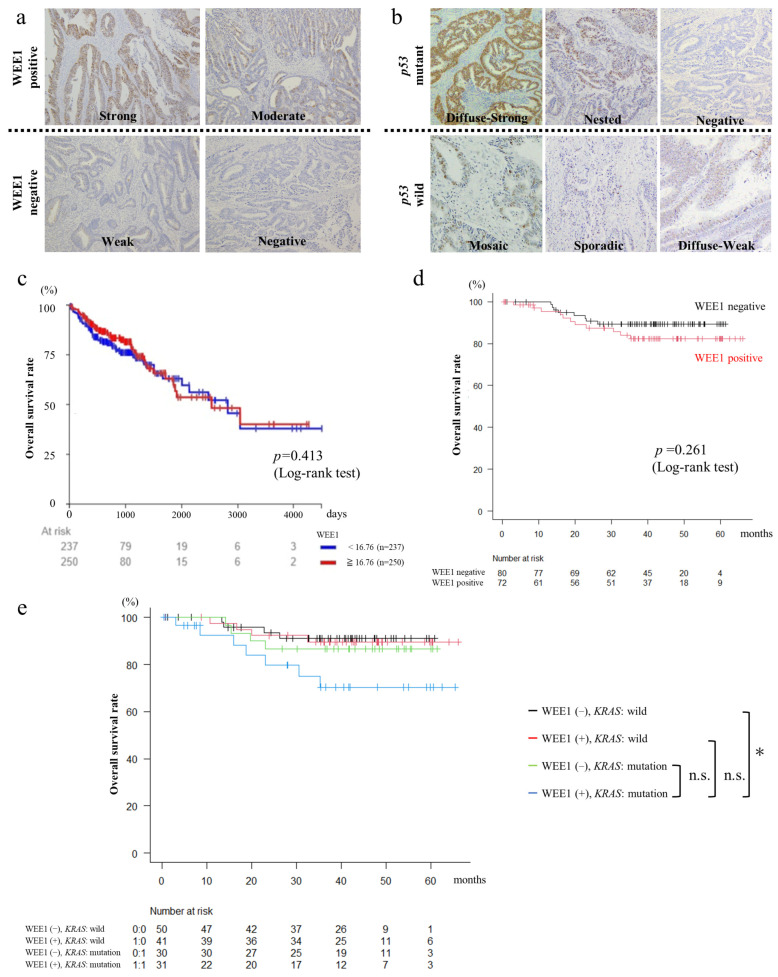
Association of WEE1 expression with the clinicopathological features and prognosis of CRC. (**a**) Immunohistochemical staining of WEE1 in surgical CRC specimens (100× magnification). Based on the staining intensity, the cells were classified as strong, moderate, weak, and negative, with strong and moderate being classified as the WEE1-positive group and weak and negative as the WEE1-negative group. (**b**) Immunohistochemical staining of *p53* in surgical CRC specimens (100× magnification). Based on the distribution of immunostained positive cells, the cells were classified as diffuse strong, nested, negative, mosaic, sporadic, and diffuse weak, with diffuse strong, nested, and negative being classified as the *p53* mutations and mosaic, sporadic, and diffuse weak as *p53* wild types. (**c**) Database analysis based on the UCSC cancer genomic browser relationship between WEE1 expression and prognosis (overall survival rate) in CRC (GDC, The Cancer Genome Atlas (TCGA), Colon Adenocarcinoma (COAD) *n* = 487). (**d**) Association between WEE1 expression and prognosis (overall survival rate) in 152 patients with surgically resected CRC at our hospital. (**e**) Comparison of the overall survival rates of patients (*n* = 152) who underwent surgery for CRC classified by WEE1 expression and *KRAS* mutation status. Log-rank test *: *p* < 0.05, n.s.: not significant.

**Figure 2 cancers-16-03136-f002:**
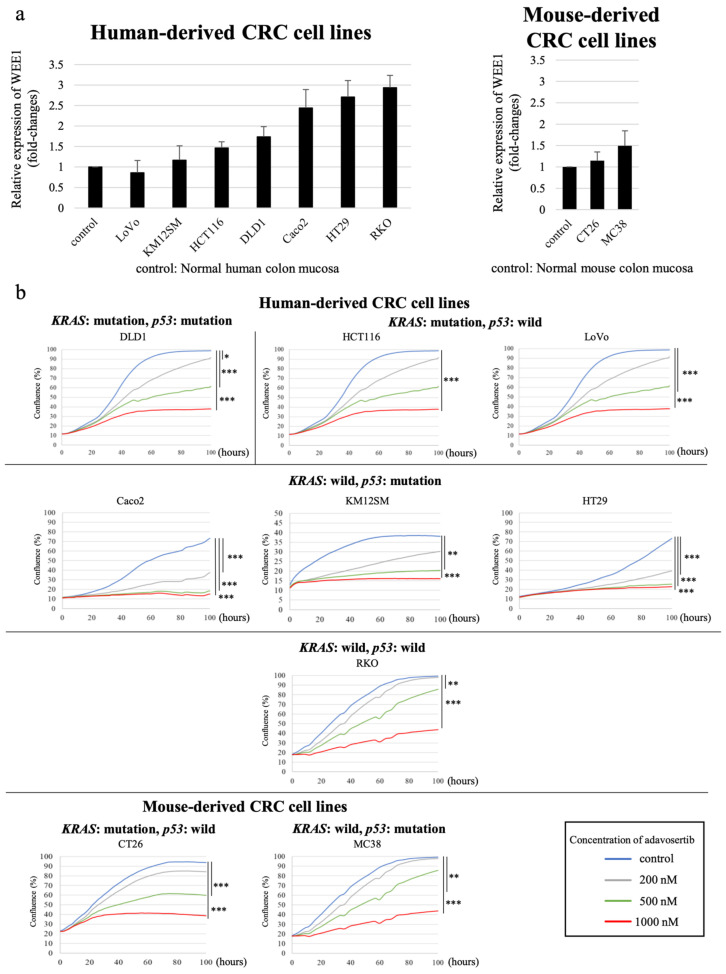
Quantification of WEE1 expression in CRC cell lines and effect of WEE1 inhibitor on cell proliferation. (**a**) Relative expression levels of WEE1 in various CRC cell lines measured using real-time PCR, where its expression level in normal colonic mucosa was set as 1. (**b**) Growth curves of various CRC cell lines after exposure to various concentrations of the WEE1 inhibitor, adavosertib (200, 500, and 1000 nM), and untreated cells (control). The mean values are plotted. One-way ANOVA followed by Dunnett’s post hoc test, *: *p* < 0.05, **: *p* < 0.01, ***: *p* < 0.001.

**Figure 3 cancers-16-03136-f003:**
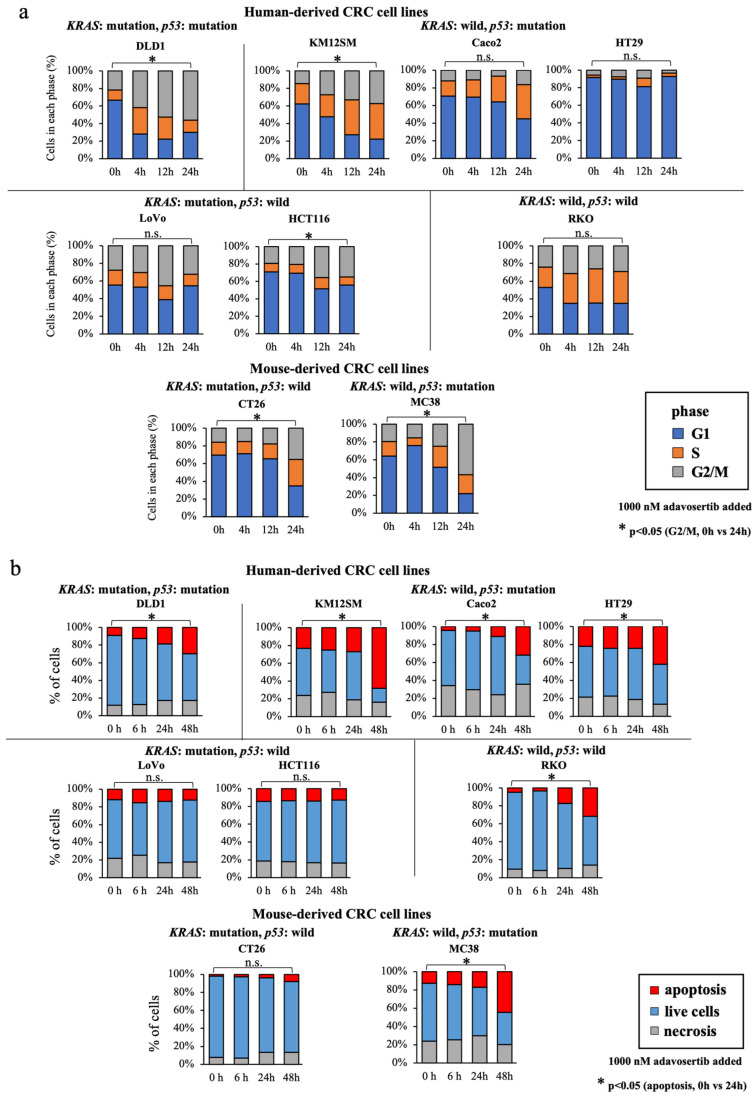
Effects of the WEE1 inhibitor on cell cycle and apoptosis induction in CRC cell lines. (**a**) Changes over time in the percentage of cells in the G1, S, and G2/M phases of the cell cycle as measured by cell cycle analysis in untreated (0 h) and various adavosertib-treated (1000 nM) CRC cell lines (at 4, 12, and 24 h). Comparison of percentage of apoptotic cells before (0 h) and 24 h after addition of adavosertib. *: *p* < 0.05, n.s.: not significant. (**b**) Changes over time in the percentage of apoptotic, viable, and necrotic cells in various untreated (0 h) and adavosertib-treated colon cancer cell lines. The cell lines were treated with 1000 nM adavosertib, and the analysis was performed 6, 24, and 48 h after treatment. Comparison of percentage of apoptotic cells before (0 h) and 48 h after the addition of adavosertib. *: *p* < 0.05, n.s.: not significant.

**Figure 4 cancers-16-03136-f004:**
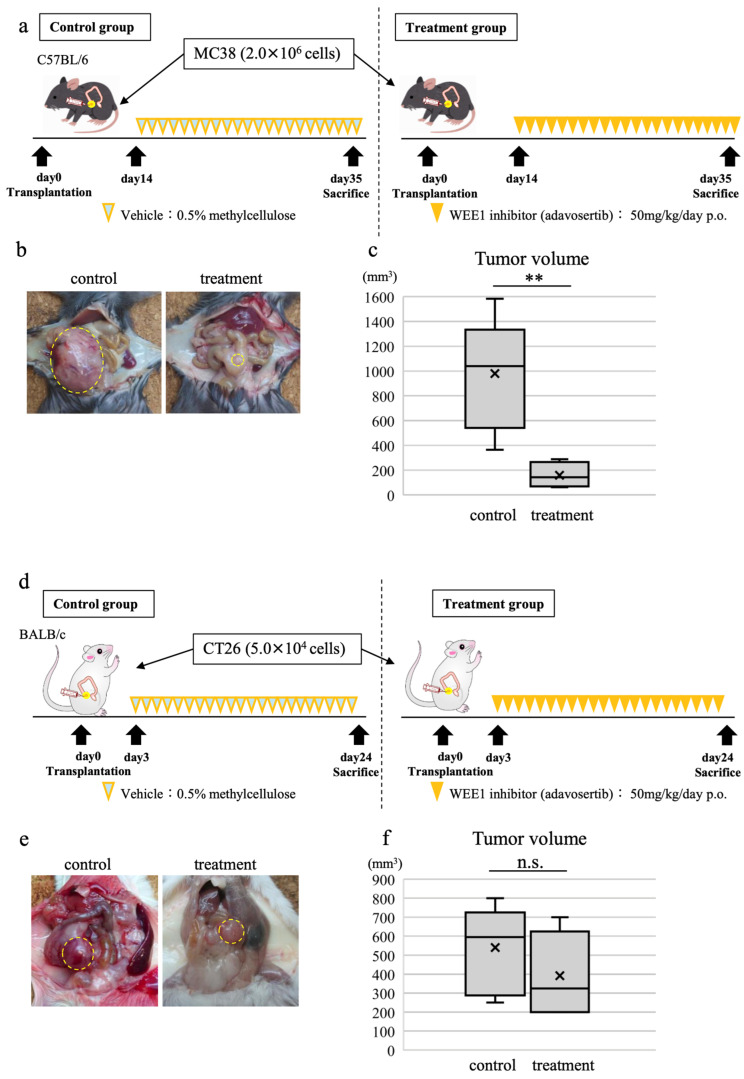
Effects of the WEE1 inhibitor, adavosertib, on tumor volume in an orthotopic transplanted mouse model of CRC. (**a**) Orthotopic transplanted mouse model of CRC using MC38: transplant design and treatment schedule. *n* = 5, control group; *n* = 4, treatment group. (**b**) Macroscopic images of mouse tumors obtained following euthanasia on day 35 post-transplantation. (**c**) Comparison of tumor volume between control and treatment groups. (**d**) Orthotopic transplanted mouse model of CRC using CT26: transplant design and treatment schedule. *n* = 6, control group; *n* = 6, treatment group. (**e**) Macroscopic images of mouse tumors obtained following euthanasia on day 24 post-transplantation. (**f**) Comparison of tumor volume between control and treatment groups. Welch’s *t*-test, **: *p* < 0.01, n.s.: not significant.

**Figure 5 cancers-16-03136-f005:**
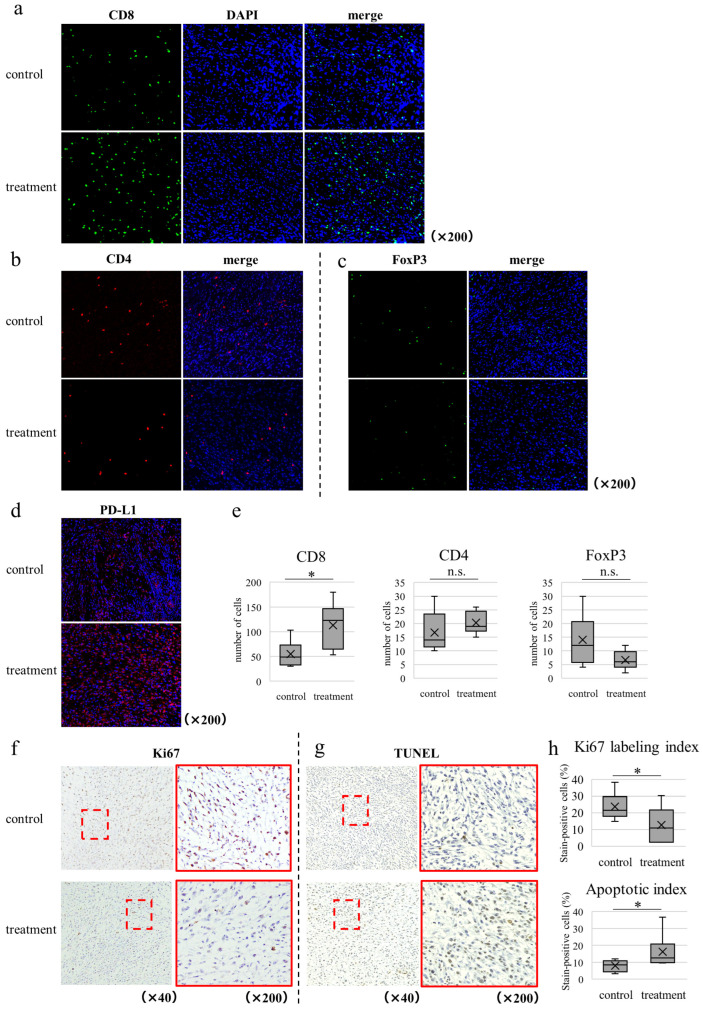
Evaluation of transplanted tumor specimens from a mouse model of CRC using MC38 by immunohistochemical staining. (**a**–**c**) Fluorescence immunostaining (200× magnification) of (**a**) CD8, (**b**) CD4, and (**c**) FoxP3. (**d**) Fluorescence immunostaining of PD-L1 (200× magnification). PD-L1: red; 4′,6-diamidino-2-phenylindole (DAPI): blue. The area around DAPI is stained because PD-L1 is expressed on the cell membrane. (**e**) Comparison of the number of CD8, CD4, and FoxP3-positive cells in the control and treatment groups. (**f**) Ki-67 immunostaining (200× magnification); (**g**) TUNEL staining (200× magnification). (**h**) Comparison of Ki-67 labeling index and percentage of TUNEL staining-positive cells between the control and treatment groups. Welch’s *t*-test, *: *p* < 0.05, n.s.: not significant, PD-L1: programmed cell death ligand 1; TUNEL: terminal deoxynucleotidyl transferase dUTP nick-end labeling.

**Figure 6 cancers-16-03136-f006:**
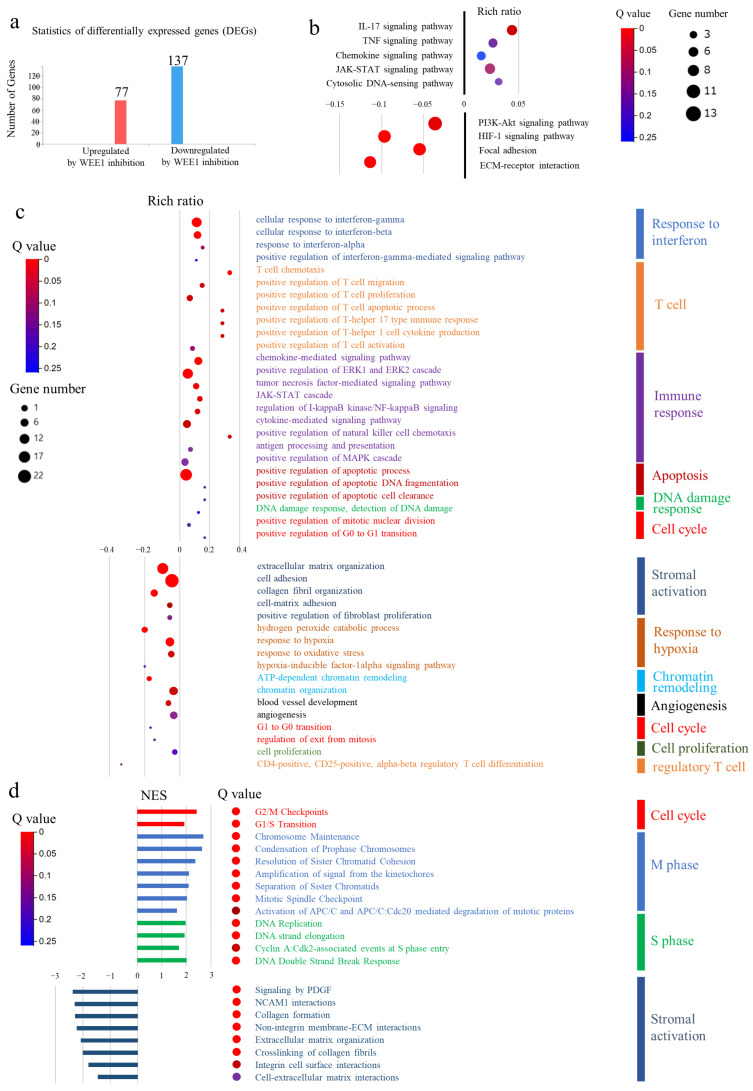
Evaluation of the gene expression profiles in transplanted tumors using RNA sequencing. (**a**) Statistics of differentially expressed genes (DEGs; *p* < 0.05 and fold change > 2). (**b**) Kyoto Encyclopedia of Genes and Genomes (KEGG) pathway analysis. (**c**) Gene Ontology (GO) enrichment analysis. (**d**) Gene Set Enrichment Analysis (GSEA) in orthotopically transplanted tumors treated with or without WEE1 inhibitor. NES: normalized enrichment score.

**Table 1 cancers-16-03136-t001:** Primers used in this study.

Target Gene	Direction	Sequence (5′–3′)
Human GAPDH	Forward	CCACCCATGGCAAATTCC
	Reverse	TGATGGGATTTCCATTGATGAC
Human WEE1	Forward	GCTTGCCCTCACAGTGGTATG
	Reverse	CCGAGGTAATCTACCCTGTCT
Mouse GAPDH	Forward	GCCTCGTCCCGTAGACAAAA
	Reverse	CCATTCTCGGCCTTGACTGT
Mouse WEE1	Forward	CCGGGGCTTGAGATACATACA
	Reverse	CAGCATTTGGGATTGAGGTTCG

GAPDH, glyceraldehyde-3-phosphate dehydrogenase.

**Table 2 cancers-16-03136-t002:** WEE1 expression and clinicopathological features.

Features		WEE1 Negative (*n* = 81)	WEE1 Positive (*n* = 71)	*p*-Value ^1^
Age (years)		68.4 ± 10.8	65.7 ± 12.8	0.157
Sex	Female	29 (35.8)	33 (46.5)	0.181
	Male	52 (64.2)	38 (53.5)	
Location	Right side colon	30 (37.0)	24 (33.8)	0.678
	Left side colon	51 (63.0)	47 (66.2)	
Histological type	tub1/2	74 (91.4)	63 (88.7)	0.588
	por/muc	7 (8.6)	8 (11.3)	
Stage	I/II	37 (45.7)	39 (54.9)	0.255
	III/IV	44 (54.3)	32 (45.1)	
T	1/2	23 (28.4)	24 (33.8)	0.472
	3/4	58 (71.6)	47 (66.2)	
N	0/1	38 (46.9)	40 (56.3)	0.246
	2/3	43 (53.1)	31 (43.7)	
M	0	65 (80.2)	60 (84.5)	0.493
	1	16 (19.8)	11 (15.5)	
V	0/1	71 (87.7)	60 (84.5)	0.575
	2/3	10 (12.3)	11 (15.5)	

^1^ The χ^2^ test and Welch’s *t*-test were conducted for categorical and continuous data, respectively. Data are presented as *n* (%). tub1, well-differentiated adenocarcinoma; tub2, moderately differentiated adenocarcinoma; por, poorly differentiated adenocarcinoma; muc, mucinous adenocarcinoma; T, tumor; N, node; M, metastasis; V, venous invasion.

**Table 3 cancers-16-03136-t003:** WEE1 expression and gene mutation status.

Gene	Status	WEE1 Negative (*n* = 81)	WEE1 Positive (*n* = 71)	*p*-Value ^1^
*KRAS*	Wild type	51 (63.0)	40 (56.3)	0.406
	Mutant	30 (37.0)	31 (43.7)	
*p53*	Wild type	12 (14.8)	7 (9.9)	0.357
	Mutant	69 (85.2)	65 (90.1)	
*BRAF*	Wild type	81 (100.0)	66 (93.0)	0.015
	Mutant	0 (0.0)	5 (7.0)	
MSI	MSS	78 (96.3)	62 (87.3)	0.041
	High MSI	3 (3.7)	9 (12.7)	

^1^ The χ^2^ test was conducted. Data are presented as *n* (%). MSI, microsatellite instability; MSS, microsatellite stability.

**Table 4 cancers-16-03136-t004:** CRC cell lines and gene mutation status.

CRC Cell Lines	*p53* Status	*KRAS* Status	*BRAF* Status	MSI/MSS
DLD1	Mutant	Mutant	Wild type	MSI
KM12SM	Mutant	Wild type	Mutant	MSI
Caco2	Mutant	Wild type	Wild type	MSS
HT29	Mutant	Wild type	Mutant	MSS
HCT116	Wild type	Mutant	Wild type	MSI
LoVo	Wild type	Mutant	Wild type	MSI
RKO	Wild type	Wild type	Mutant	MSI
CT26	Wild type	Mutant	Wild type	MSS
MC38	Mutant	Wild type	Wild type	MSI

CRC, colorectal cancer; MSI, microsatellite instability; MSS, microsatellite stability.

## Data Availability

The original contributions presented in this study are included in the article/Appendix A; further inquiries can be directed to the corresponding author.

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
