# Peer review of "WEE1 Inhibitor Adavosertib Exerts Antitumor Effects on Colorectal Cancer, Especially in Cases with p53 Mutations"

_cancers, 2024, doi:10.3390/cancers16183136_

Round 1
Reviewer 1 Report
Comments and Suggestions for Authors
In this study, Ariyoshi et al. evaluated the effect of inhibiting WEE1, a protein kinase that regulates the G2/M checkpoint of the cell cycle, in human colon cancer cells lines and CRC mouse models. From analyzing CRC tumor samples, the authors found that WEE1 positivity is associated with BRAF mutation and MSI-H status, and that patients with WEE1-positive and KRAS mutant CRC have worse overall survival. In CRC cell lines, the authors looked at the effect of adavosertib on cell proliferation, cell cycle and apoptosis. In orthotopic CRC mouse models, they looked at the effect of adavosertib on tumor size and markers of cell proliferation, apoptosis, and tumor immunity.
Overall, the manuscript is well written, and the results are interesting, well presented, and mostly clear.
Specific comments:
1. The results of the qPCR (Fig 2a) do not quite match the description in the text (lines 313-315) and so the text should be revised for accuracy. The authors need to show statistical analysis to determine significant differences in expression between the various cell lines. Also, what is the source of control RNA = normal human colon mucosa?
2. Statistical analysis is also lacking for cell cycle and apoptosis experiments (Fig 3a-b) and should be provided. Also, HT-29, which has p53 mutation, does not appear to fit the trend of increased % G2/M phase and increased apoptosis (Fig 3a-b), so the authors should comment on that.
3. Revise lines 479-480 in the Discussion for accuracy since ICIs have shown efficacy in a subset of CRC, and are part of standard treatment for advanced or metastatic MSI-H/dMMR tumors.
Reviewer 2 Report
Comments and Suggestions for Authors
Article elaborating the role of WEE1 Inhibitor (Adavosertib) Exerting the Antitumor Effects on Colorectal Cancer. This is a well-written manuscript with significant implications in the translational aspect. This research article follows the hypothesis of the work. However, a few things must be addressed before it is ready for acceptance. They are as follows:
1. Fig 5 a-d, all IF figures must be adequately zoomed and shown in this manuscript with better resolution.
2. Authors must add a model depicting the take-home message from this work and discuss it thoroughly by adding a few lines in the conclusion part.
3. It has been shown that WEE1 inhibition induces glutamine addiction (PMID: 31919076), while it has also been shown how oncogenic KRAS regulates cell cycle-regulated checkpoints and glutamine metabolism (PMID: 26682255).
It is speculative that Adavosertib might play a definitive role in cancer metabolism, more prominently in glutamine metabolism. This would be one of the future directions as a follow-up of this research. Authors must add a few lines discussing this in the introduction by adding mentioned relevant works.
Round 2
Reviewer 2 Report
Comments and Suggestions for Authors
All concerns addressed and ready for acceptance.